# Organotypic Culture of Acinar Cells for the Study of Pancreatic Cancer Initiation

**DOI:** 10.3390/cancers12092606

**Published:** 2020-09-12

**Authors:** Carlotta Paoli, Alessandro Carrer

**Affiliations:** 1Veneto Institute of Molecular Medicine (VIMM), 35129 Padova, Italy; carlotta330@gmail.com; 2Department of Biology, University of Padova, 35129 Padova, Italy

**Keywords:** pancreas, pancreatic cancer, acinar-to-ductal metaplasia, organotypic culture, KRAS, metabolic reprogramming, epigenetic reprogramming

## Abstract

**Simple Summary:**

Pancreatic Cancer is a deadly disease, with a dismal prognosis. A better understanding of the molecular alterations that cause the malignant transformation of pancreatic epithelial cells is pivotal to curtail disease incidence and detect the disease early when it can be surgically resected. The culture of pancreatic pre-malignant cells is technically challenging, but a great tool for the study of tumor evolution and early oncogenic alterations. Here, we will describe the isolation of pancreatic acinar cells and its value for the study of tumor initiation, from a technical and historical perspective.

**Abstract:**

The carcinogenesis of pancreatic ductal adenocarcinoma (PDA) progresses according to multi-step evolution, whereby the disease acquires increasingly aggressive pathological features. On the other hand, disease inception is poorly investigated. Decoding the cascade of events that leads to oncogenic transformation is crucial to design strategies for early diagnosis as well as to tackle tumor onset. Lineage-tracing experiments demonstrated that pancreatic cancerous lesions originate from acinar cells, a highly specialized cell type in the pancreatic epithelium. Primary acinar cells can survive in vitro as organoid-like 3D spheroids, which can transdifferentiate into cells with a clear ductal morphology in response to different cell- and non-cell-autonomous stimuli. This event, termed acinar-to-ductal metaplasia, recapitulates the histological and molecular features of disease initiation. Here, we will discuss the isolation and culture of primary pancreatic acinar cells, providing a historical and technical perspective. The impact of pancreatic cancer research will also be debated. In particular, we will dissect the roles of transcriptional, epigenetic, and metabolic reprogramming for tumor initiation and we will show how that can be modeled using ex vivo acinar cell cultures. Finally, mechanisms of PDA initiation described using organotypical cultures will be reviewed.

## 1. Introduction

Pancreatic adenocarcinoma (PDA) is the most common form of pancreatic cancer, a disease that poses significant therapeutic challenges and accounts for more than 400,000 deaths worldwide [1]. The extremely poor survival rate of 9% with a five-year survival rate makes PDA the most lethal among major forms of cancer [2]. PDA remains asymptomatic for years and manifests at very late stages, often making the tumor unresectable [2,3]. A significant improvement of patient prognosis comes with earlier diagnoses, which are mostly anecdotal due to the poor understanding of disease initiation and the lack of biomarkers for early detection [4]. 

PDA progresses through a multi-step mechanism, where tumor formation is preceded by the formation of non-invasive pre-neoplastic lesions [5,6]. Genetic, epigenetic, and microenvironmental alterations are necessary for the formation of pre-malignant foci as well as for their progression to frank carcinoma. Understanding the cascade of events that leads to oncogenic transformation could illuminate means to prevent the onset of this deadly disease, or to enhance early diagnosis [4].

Dysplastic lesions are characterized by a distinctive duct-like morphology, which has led to the assumption that PDA arises from the expansion of mutated ductal cells. However, this notion has been increasingly disputed by several pieces of evidence since the early 1990s. More recently, a combination of lineage tracing experiments, genomic analyses of human specimens and in vitro examinations led to the notion that most pancreatic tumors originate from post-mitotic exocrine acinar cells (discussed below and in [7]).

This review will focus on early steps of pancreatic carcinogenesis and will describe the usage of a three-dimensional, organotypic culture of primary pancreatic epithelial cells for the interrogation of molecular alterations causing neoplastic transformation of acinar cells. The value of this approach as an ex vivo model of tumor initiation will be discussed and notable results obtained using this in vitro tool will be presented. 

## 2. Anatomy and Physiology of the Pancreas

Located in the retroperitoneum, the pancreas extends from the C-shaped curve of the duodenum (head), passes behind the stomach (body), and finishes at the hilum of the spleen (tail). Anatomical placement and gross anatomy of the organ are illustrated in Figure 1A.

During development, pancreatic specification starts from two early buds emerging from the gut tube and is controlled by the timely expression of a set of master transcription factors, including Pancreatic Transcription Factor 1a (*Ptf1a*), Pancreatic-Duodenal Homeobox 1 (*Pdx1*), and Sry-related HMG box 9 (*Sox9*). The developmentally mature pancreas is composed by a diverse arrangement of epithelial cells (acinar, ductal, the array of endocrine cells) and stromal cells [6], as schematically shown in Figure 1B. All epithelial cell types retain an elevated degree of plasticity in adult life and can transdifferentiate, a feature with significant pathophysiological and therapeutic ramifications [8]. 

Part of the digestive system, the pancreas is best known as the insulin-secreting gland. Insulin is produced by Langerhans’ islets, that are discrete groups of specified cells scattered across the organ and also produce glucagon and somatostatin. However, the pancreas is an extremely miscellaneous tissue (Figure 1B) where the large majority of its parenchyma functions as an exocrine accessory digestive organ which synthesizes, stores and secretes a wide set of digestive enzymes [9]. The main pancreatic duct runs longitudinally through the extension of the organ, and branches in a tree of smaller conduits (Figure 1A) that are responsible to collect the digestive enzymes produced in the lobular units of the pancreas. Secretory units are organized in packed serous acinar glands, or acini, which form an anastomosis with the tubular network (Figure 1C). Each acinus is composed of post-mitotic epithelial cells, termed acinar cells (AC), and a few intercalated ductal cells that form the intra-acinar portion of the ductal system, along with one centroacinar cell that exhibits mixed traits [6]. 

ACs are polarized, pyramidal-shaped cells containing numerous acidophilic granules near the apical side. Those granules contain inactive proteases, which are activated and released into the tubular network upon activation. Exocytosis is promoted by elevation of circulating levels of cholecystokinin (CCK), gut-derived Secretin, or by direct vagal stimulation [9,10]. Early immunostaining examinations and recent analyses of individual cell transcriptomics (sc/snRNA-Seq) indicate the existence of distinct subpopulations of acinar cells that might reflect regional and/or functional differences [11,12]. The biological significance of the diversity in the exocrine compartment is difficult to justify [13]. In fact, the herd of ACs is histologically and morphologically indistinguishable. Some subsets may possess superior replicative capacity to support tissue homeostasis and repair [11]. However, ACs are long-living and the turnover of the pancreatic epithelium is limited. Thus, the presence of pluripotent-like cells in the adult pancreas is still debated [13].

## 3. Pancreatic Carcinogenesis

PDA tumors are frequently located in the head of the pancreas, and overhaul pancreatic tissue architecture, with extensive stromal deposition and over-representation of duct-like structures [6]. Endocrine and exocrine functions are impaired early on, but remain competent for a long time [3]. The reconstruction of tumor cells’ genomic history indicates that at least eight years span between the first oncogenic event and the detection of an overt PDA mass [14]. 

This clinically silent incubation period implies a typical late manifestation, which has led to a gap in the study of early stages of pancreatic carcinogenesis. 

Genomic and histopathological analyses of either autoptic pancreata or early stage lesions or aggressive tumors only partially illuminate the transformative molecular events that lead to PDA [14,15,16,17]. In the early 2000s, the development of autochthonous mouse models of pancreatic tumorigenesis allowed the monitoring of disease progression since its inception [18,19], but only recent lineage-tracing experiments demonstrated that cancerous foci predominantly originate from metaplastic acinar cells, which undergo ductal transdifferentiation in response to environmental or oncogenic stress [20]. This tumor-initiating event is termed acinar-to-ductal metaplasia (ADM), and is normally a physiologic, reversible process that supports tissue repair. Temporary metaplasia occurs during pancreatic regeneration after injury. In mice, drug-induced inflammation causes extensive loss of acinar cells and drives temporary ADM in surviving cells to restore normal tissue histology after few days [21]. However, oncogenic inputs (both cell- and non-cell-autonomous) make ADM irreversible so that metaplastic cells engage a step-wise process that leads to the formation of low- and high-grade in situ dysplasia and eventually to metastatic carcinoma (Figure 2A) [5,22]. This review describes an in vitro model of ADM to interrogate the factors that unlock acinar cell plasticity.

A variety of dysplastic precursors have been classified according to histopathological characteristics. Pancreatic intraepithelial neoplasia (PanIN) is the most commonly observed, and histological analysis of human specimens often reveals areas of acinar-to-ductal metaplasia in close proximity to low-grade PanIN foci [23,24], suggesting that the latter could in fact arise from the expansion of metaplastic cells. While the acinar origin of PanINs is accepted [5,7], their disproportional representation in autochthonous models makes the origin of human PDAs still debated. Rarer, more benign cyst-like lesions are observed in humans and they might also evolve to PDA in some cases [2,3]. The acinar cell origin of these dysplastic lesions is less likely.

Mutations in the coding sequence of the *KRAS* oncogene are nearly omnipresent in human PDA samples, but are also commonly observed in pre-neoplastic lesions [5]. Bioinformatic deconvolution of whole-genome analyses traced *KRAS* mutations very early in the oncogenic progression [14]. Strikingly, mice expressing the mutated form of KRAS (*Kras^G12D^*) in the pancreatic epithelium (*Pdx1-Cre;LSL-Kras^G12D^*-“KC mice”) spontaneously develop widespread ADM, which lead to the appearance of several PanIN foci by few weeks of age [18]. Of note, no other dysplastic lesions are typically observed in KC mice [18]. Current research shows that activating mutations of *KRAS* can exploit acinar cell plasticity to trigger ADM, which eventually progresses to form cancerous lesions upon additional genetic and epigenetic alterations [22,25,26]. PDA can originate from other pancreatic epithelial cells (i.e., ductal cells), which are however more refractory to the sole effect of oncogenic KRAS and may require the synergism of different oncogenes, or alternatively give rise to less common, more benign, precursor lesions [20,27,28].

Understanding the role of acinar cells in disease is of utmost importance due to their abundance in the parenchyma and their marked vulnerability to oncogenic insults.

The pancreas is a composite tissue, also housing a resident stromal population (pancreatic stellate cells), connective tissue that encapsulates the acini and the entire pancreas, adipocytes, and endothelial cells. The stromal compartment is further enriched as the disease progresses and can exert both tumor-promoting and -opposing functions [29,30,31,32,33]. Studying the communication between stromal cells and the PDA cell of origin (acinar cells) is important to decipher the cellular mechanisms of disease onset.

## 4. Organotypic (3D) Acinar Cell Culture

Biochemical examination of acinar cells in vitro is essential to elucidate the cellular mechanisms of pancreatic carcinogenesis [34]. The first example of acinar cell isolation was reported by Amsterdam and Jamieson in 1972. In their approach, pancreata harvested from a guinea pig were digested with a collagenase–hyaluronidase mixture and cultured ex vivo for few days [35]. The inability to cultivate isolated cells for long periods of time suggested the preservation of acinar cells’ post-mitotic differentiation in culture. However, appropriate phenotypic characterization was performed only several years later. Immunostaining revealed that, while initially composed entirely by *Amylase*-expressing cells (denoting protease-secreting cells), cultures were enriched in cytokeratins-positive cells (denoting ductal phenotype) four days after isolation [34,36] (Figure 2B). 

At first, the presence of duct-like cells was attributed to inevitable contamination (and subsequent expansion) of cells endowed with higher proliferative capacity [7]. This supported the narrative that PDA originates from ductal cells. However, in a landmark study, Means et al. showed that differentiated exocrine cells from adult mouse pancreas undergo transdifferentiation into duct-like cells [37]. This process was immediately linked to acinar-to-ductal metaplasia, a phenomenon required for pancreatic tissue repair that had also been observed in scattered association with low-grade PanIN lesions in PDA patients [23,37]. Time course analysis revealed that acinar cells progressively acquire the expression of ductal-specific genes, while downregulating genes that typify the exocrine function of the pancreas (e.g., proteases like Carboxypeptidase-A1 (Cpa1) or Amylase-2 [Amy2]) (Figure 2C). In the original study, Transforming Growth Factor α (TGFα) was used to trigger ADM ex vivo. Although TGFα is still being used, later studies allowed the identification of other factors equally able to induce metaplasia (see below).

The protocol for the isolation and the culture of acinar cells has been optimized by several laboratories over the past decades [38,39,40,41], but still hinges on a light enzymatic digestion of minced pancreata to preserve the tri-dimensional organization of the acini. Digestion time is variable [39] and results in a heterogeneous mixture of microscopic tissue explants. Cocktails of Ser/Thr proteases inhibitors are added in order to prevent autodigestion of the explants (due to stress-induced release of proteases from acinar cells). A gradient sedimentation step enriches the preparation for acini, which sink after slow centrifugation, although that does not impede the accidental co-isolation of other cell populations. Finally, cell suspension is resuspended in Waymouth’s complete medium, or alternatively embedded in a collagen I matrix or Matrigel^®^. The short-term suspension culture of dispersed acini is preferred to study the biochemistry and pathophysiology of the exocrine pancreas [42]. A matrix scaffold can extend the in vitro culture period to up to 10 days and allows morphological and immunostaining analyses, but induces spontaneous ADM [38]. In particular, Matrigel^®^ contains a population of growth factors that varies considerably across batches and can impact cell identity, as illustrated below. 

This procedure has several caveats. One hindering element is the intrinsic exposure to lytic enzymes, proteases, and lipases. To overcome this problem, Bläuer and colleagues have optimized a culture model in which pancreatic tissue explants are maintained at the gas-liquid interphase on a perforated membrane, leaving the acinar cells to migrate out of the explant and form a monolayer on the membrane surface without the addition of collagenolytic or proteolytic enzymes [42]. More recently, the application of fluorescence-activated cell sorting (FACS) to isolate different pancreatic cell populations has been tested. Even if this method is efficient to obtain a pure population of acinar cells, it requires specific labels, the digestion to single cell suspensions, and does not permit a long-term culture of purified cells [38]. Still, FACS-sorted cells have successfully been used for molecular biology analysis, including the epigenomic characterization of different pancreatic lineages [43]. 

## 5. 3D Culture of ACs to Study Pancreatic Cancer Initiation

The culture of dispersed acini has been deployed for the study of exocrine pathologies and the dynamics of enzyme secretion during food digestion (reviewed elsewhere [10]). However, in light of the observation that PDA often originates from acinar cells that underwent ADM [20], it has been adopted by cancer researchers to interrogate the molecular pathways leading to transdifferentiation and initiation of oncogenesis. 

The lack of non-transformed acinar cell lines makes the explant and culture of primary acini the sole approach to directly investigate the biology of tumor initiation in a cell-based system. Of note, mesenteric organs (including the pancreas) are poorly accessible and biopsies are limited in clinical practice, which causes scarce availability of human pre-neoplastic acinar cells. In contrast, primary murine cultures are widely deployed in molecular biology laboratories and offer several advantages, notwithstanding potential biological differences, which are discussed elsewhere [44]. 

Murine ex vivo cultures are excellent surrogates for in vivo models. When embedded in a 3D matrix (typically, collagen or Matrigel^®^), freshly isolated acinar cells organize in grape-like structures (acini), easily recognizable using an optic microscope. As outlined below, both cell-intrinsic and -extrinsic factors can trigger ductal transdifferentiation (Figure 2B). Ex vivo, ADM is often initiated by addition of recombinant TGFα or EGF, although that is not necessary when using growth factor-rich matrices (such as Matrigel^®^). During this process, acinar cells change morphology, but also lose their secretory function and markedly alter their gene expression profile, progressively increasing the expression of duct-specific genes, and reducing the levels of genes encoding for acinar proteases (i.e., Carboxypeptidase-A1 (*CPA1*) or Amylase-*2* (*AMY2*) (Figure 2C). After a few days (variable), an elementary morphological examination reveals the presence of several duct-like, flat, circular structures, not originally present in the plate. Although a level of plasticity is innate in acinar cells, acini-to-duct conversion can be accelerated or suppressed by multiple factors that ultimately also either promote or inhibit pancreatic tumor initiation in vivo. The impact of genetic mutations, growth factor stimulation, gene silencing/overexpression or drug administration can be objectively quantified ex vivo: by counting the numbers of duct-like structures or quantifying the expression of ductal-specific markers (e.g., Cytokeratin-19, *KRT19*) after a few days in culture, investigators evaluate the propensity of acini to undergo ADM, which correlates with tumor initiation capacity. The impact of drug administration, gene targeting, or specific genetic mutations can be examined, as schematically outlined in Figure 3A,B.

Next, we will review how the organotypic culture of primary pancreatic acini has led to findings that improved our understanding of pancreatic tumor onset and predisposition. 

## 6. Genetic, Epigenetic and Metabolic Alterations Guide Ex-Vivo ADM and Support Tumor Initiation In Vivo

Genetically-engineered mice expressing oncogenic *Kras* in the pancreatic epithelium show diffuse metaplastic acini [18]. Together with the compelling prevalence of its mutated form in human tumor samples, this pointed to oncogenic KRAS as the key driver of PDA carcinogenesis [26]. Due to the impact of KRAS being largely cell type-dependent [45,46,47], the analysis of oncogenic mutations in non-transformed acinar cells is critical to characterize the precise mechanism of tumor initiation and possibly identify targetable processes to prevent cancer onset.

Acinar cell-restricted expression of *Kras^G12D^* induces the formation of duct-like circular structures in 3D cultures of acinar cell explants [48]. Although cell transdifferentiation occurs spontaneously in these settings, single-allelic *Kras* mutations significantly accelerate ADM and enhance the expression of ductal markers like *Sox9* and *Krt19*, while suppressing canonical acinar genes (*Mist1*, Amylase) [48,49]. This effectively illustrates the value of acinar cell cultures to study PDA initiation: the tumorigenic potential of KRAS^G12D^ can be quantified either by morphological examination of the acini-turned-ducts or by quantitative PCR (Figure 3B,C). 

In vitro, the activation of mutant *Kras* can be timely controlled, for example through the administration of *Cre*-encoding viral vectors [50] or viral constructs guiding site-specific recombination [51]. Notably, acinar cells are efficiently infected by all major delivery vectors, adenoviruses [50], adeno-associated viruses [52], and lentiviruses [53]. Similarly, vector mediated-genetic manipulations can be applied to any gene of interest. Additionally, this organism-free set up permits in-depth biochemical analysis of cell-intrinsic signaling, metabolic and epigenetic reprogramming which occur downstream of, or cooperate with oncogenic KRAS to drive acinar-to-ductal metaplasia as discussed hereafter.

### 6.1. The Usual Suspects: Tumor-Associated Signaling Pathways Induce ADM in Organotypic Culture

Signaling pathways activated by oncogenes or common tumor-associated cytokines have been studied in ex vivo ADM. The classical mitogen-activated protein kinase (MAPK) cascade (also: ERK pathway) is a prominent signaling effector pathway, which mediates many mitogenic inputs [54]. The MAPK cascade is constituted by sequential protein phosphorylations that are initiated by RAS proteins when tyrosine kinase receptors (such as the epidermal growth factor receptor (EGFR), which recognizes the proto-typical ADM-inducers TGFα and EGF [55]) bind their ligands or when RAS is constitutively activated by mutagenesis, and culminates with the activation of multiple transcription factors that have major impact on cell proliferation and behavior [54]. 

MAPKs drive irreversible ADM. Multiple tyrosine receptors’ ligands induce metaplasia both ex vivo (human and murine acinar explants; [37,41]) and in vivo in multiple transgenic models [56,57,58]. On the other hand, *Egfr* ablation or pharmacological targeting suppress tumor initiation in murine models, while minimally impacting tumor progression [59,60]. Importantly, treatment with the EGFR Kinase inhibitor Erlotinib blocks acinar-to-ductal metaplasia in *Kras^G12D^*-expressing acinar cells ex vivo [59]. Similarly, silencing Mapk Erk Kinase (MEK) isoforms prevents cerulein-promoted tumor onset [61]. Finally, multiple reports show that pharmacological targeting of MEK kinases inhibits transdifferentiation of isolated acini [48,61,62]. The data obtained from isolated acini clearly highlight the role of the MAPK signaling pathway in the induction and maintenance of acinar cell metaplasia.

On the other side, a wider spectrum of signaling pathways may synergize to promote ADM. Protein Kinase B (PKB/AKT) is a major signaling node that broadly regulates cellular and systemic functions [63]. Multiple reports show that selective AKT inhibition suppresses ADM in isolated acini in vitro [64,65,66] while also show that AKT is activated by a variety of upstream regulators. 3-Phosphoinositide-Dependent protein Kinase 1 (*Pdk1*) ablation in KC mice impedes both AKT phosphorylation in PanIN cells and tumor initiation [65]. Similarly, Western blot analysis of dispersed acini shows that haplodeficiency of Unfolded Protein Response Regulator 78 (GRP78) reduces both AKT phosphorylation and the formation of duct-like structures at end point [67]. Notably, AKT is activated by EGF and TGFα in both primary acinar cell cultures [68] and PDA cells [64], among several other factors such as Insulin and Insulin-Like Growth Factor [64,69,70]. Together, this identifies the KRAS/PI3K/AKT axis as a critical, targetable mechanism of PDA initiation that integrates cell-intrinsic and -extrinsic signals.

A number of signaling pathways have been linked to the acinar/ductal switch of cellular identity, as extensively outlined elsewhere [22]. Interestingly, several inflammatory nodes are upregulated in advanced PDA and also promote ADM, both in vitro and in vivo (e.g., Signal Transducer and activator of transcription 3, STAT3 [71], Nuclear Factor of Activated T cells, NFATc1 [72], Intercellular Adhesion Molecule 1, ICAM1 [73], and Transforming Growth Factor beta, TGFβ [62], among others). This highlights the contribution of local inflammation to pancreatic metaplasia and tumor initiation.

Consistent with its physiological function in tissue regeneration, acinar-to-ductal metaplasia is supported in part by the re-activation of developmental pathways including the Wnt/β-catenin pathway, which is known to regulate embryonic acinar development [74]. β-catenin stabilization is indeed required for pancreatic regeneration after acinar cell loss [75], and Wnt signaling progressively increases during pancreatic carcinogenesis while acini isolated from β-catenin-deficient mice do not undergo metaplasia ex vivo [76,77]. This shows the tumor-promoting function of physiological Wnt signals.

Collectively, these findings illustrate that organotypic cultures of primary acini are a powerful set up to interrogate the biochemistry of the tumor-initiating transdifferentiation of pancreatic epithelial cells and reveal a tangled signaling nexus, finely regulated during tissue regeneration but irreversibly compromised at the onset of carcinogenesis. 

### 6.2. Master Transcription Factors Shape Cell Identity: Discerning Critical Hubs Using Primary Acinar Cultures

All epithelial lineages in the adult pancreas originate from common developmental precursors (see above and [8]). After injury, acinar cells activate developmental programs to de-differentiate into facultative progenitor-like cells, which can then specify into ductal cells to permit tissue regeneration [78,79]. Data suggest that aberrant activation of developmental transcription factors (TFs) contributes to the inception of pancreatic cancer. The same dynamics also occurs in ex vivo cultures [40]. 

The SRY-related HMG-box (SOX) family is composed by 20 TFs that regulate cell fate determination [80]. Interestingly, the expression of *Sox17* characterizes pancreatobiliary progenitors which represent a cellular intermediate in the metaplastic conversion of acinar cells to duct-like cells [81,82]. However, it is not clear whether this population may also be distinctively present among cultured metaplastic acini. Yet, forcing *Sox17* expression in acinar cells leads to widespread ductal metaplasia [81]. In line with this, the overexpression of the pancreatic progenitor cell identifier *Sox9* also drives extensive ADM in adult mice [20]. 

Pancreatic and Duodenal Homeobox 1 (PDX1) marks embryonic progenitors that eventually give origin to all pancreatic epithelial cells [83]. Consistently, *Pdx1* levels increase during ADM [40,83], while its prolonged expression in acinar cells causes metaplasia [84].

In contrast, developmental regulators that dictate acinar specification markedly restrain cell identity during tissue injury/regeneration and oppose *Kras^G12D^*-induced tumorigenesis [79]. This is epitomized by the nuclear receptor *Nr5a2*, the deletion of which accelerates ductal transdifferentiation of Matrigel-embedded acini [85], exacerbates ADM and enhances PanIN formation after induction of pancreatitis [85,86]. Transcriptomic analysis in *Nr5a2*-haplodeficient organoid cultures of primary acinar cells showed a marked rewiring of gene transcription along with genome-wide repositioning of Nr5a2 and its co-activators [86]. Other transcription factors (*Ptf1a*, *Mist1*, *Gata6*, and *Rpbjl*) can exert a comprehensive control of the transcriptional profile to preserve acinar differentiation [49,87].

Altogether these findings support the paradigm that pancreatic cell identity is orchestrated by few dominant transcriptional hubs. This implies that a small number of players might regulate cell plasticity, transdifferentiation, and tumor initiation, which allow widespread transcriptional rewiring. The parallelism with the reprogramming of inducible pluripotent stem cells (iPSCs) is evident [88]. Indeed, Kruppel-Like Factor 4 (KLF4) is one of the “Yamanaka factors”, a mighty stemness-inducing cocktail of transcription factors [88]. *Klf4* overexpression in isolated acinar cells promotes acinar-to-ductal metaplasia whereas its genetic ablation attenuates the formation of duct-like structures [89]. Similarly, KLF4 supports KRAS^G12D^-induced tumor onset in mouse [89]. The conclusion, although not directly tested, is that altered expression of *Klf4* triggers extensive transcriptional reprogramming that supports the acini-to-ducts phenotypic switch. 

A comprehensive review of transcriptional regulators of pancreatic cell identity in development and disease can be found elsewhere [22] and schematically shown in Figure 4A. In spite of, in vitro observations on cultured acini substantiate the idea that ADM is guided by broad, genome-wide alterations of gene expression that force a dramatic switch of cell identity, function, and phenotype. Broad changes in a cell’s expression profile are typically guided by epigenetic reprogramming [90].

### 6.3. Epigenetic Regulators Impose Cell Identity and Guide Phenotypic Switch: Epigenetics of Acini-to-Ducts Transition in Primary Cultures

Epigenetic marks are post-translational modifications of DNA or histones that regulate higher order chromatin organization to control DNA accessibility, transcription factor positioning and cell transcriptome. The epigenetic status ultimately imposes cell identity during developmental specification but also in adult-life plasticity [88]. Not surprisingly, acinar cells undergo significant epigenetic reprogramming during ADM (Figure 4B).

In fact, TGFα-overexpressing pancreatic epithelial explants exhibit increased histone H3 global acetylation, while also forming significantly more ducts when embedded in collagen [91]. Histone H4 acetylation is elevated during ex vivo ADM in *Kras^G12D^*-expressing acini [64], an effect also observed during tumor development in KC mice [64]. Acetylated histones determine a more relaxed chromatin state, making the cells transcriptionally more dynamic. In contrast, repressive histone modifications like H3K27me3 and H2AK119ub, are increased at loci encoding acinar cell fate genes during ex vivo ADM [92,93]. These data support the notion that large-scale epigenetic reprogramming occurs during the transition to duct-like cells in order to selectively and collectively turn off acinar specification genes. 

Epigenetic rearrangements alter transcription factors’ access to the chromatin and dictate the accessibility of binding sites, influencing their transcriptional output. Brahma related gene 1 (*BRG1*) is a component of the SWI/SNF chromatin remodeling complexes and is frequently inactivated in PDA [94]. Chromatin Immunoprecipitation (ChIP) of isolated acini revealed that BRG1 binds to the *Sox9* promoter and facilitate PDX1 recruitment through a local change of chromatin conformation [95]. Genome-wide analysis (ChIP-Seq and ATAC-Seq) in sorted acinar cells revealed that another SWI/SNF subunit, ARID1A, remodels nuclear architecture so to expose binding motifs of acinar cell-specific TFs (RPBJ, PDX1, NR5A2, PTF1). Accordingly, silencing of *Arid1a* restrains DNA accessibility during oncogene-induced ADM [43].

During tissue regeneration in mice, histone methylation temporarily increases to facilitate the recruitment of Nuclear Factor Activated in T cells, NFATc1, a transcriptional activator, at the promoter of pro-metaplastic genes, including *Sox9* [72]. ChIP analysis on primary acini showed elevation of Histone H3 Lysine 4 tri-methylation (H3K4me3, active chromatin mark) at the *Sox9* locus that aligns with the recruitment of NFATc1 and RNA polymerase II. Interestingly, *Egfr*-deficient cells show neither H3K4me3 increase nor NFATc1 recruitment, suggesting that epigenetic reprogramming at the *Sox9* locus is mediated by ADM-inducing inputs [72]. This set of experiments highlights the importance of a primary cell system to interrogate site-specific differences of epigenetic marks, as the application of ChIP-based approaches to in vivo models would be technically challenging and results would be difficult to interpret considering the cellular heterogeneity of the pancreas.

Collectively, chromatin reshaping occurs in metaplastic cells and synergizes with master transcriptional regulators to render ADM irreversible. Of note, histone remodeling genes are among the most mutated in PDA, which clearly denotes their significance for disease progression [94]. 

### 6.4. Rewiring of Cellular Metabolism Supports Ex Vivo ADM through the Production of Anabolic Intermediates

Epigenetic reprogramming is often intertwined with the rewiring of metabolic processes, in that availability of metabolites impacts the activity of several chromatin remodeling enzymes (reviewed in [96,97]). 

Histone lysine acetylation, which marks regions of “open” chromatin, is established by a pack of Histone Acetyl Transferase (HAT) enzymes. HATs transfer an acetyl moiety from a universal donor, acetyl-Coenzyme A (acetyl-CoA) that also happens to be a pivotal metabolic intermediate. Ex vivo cultures of acinar cells demonstrated that oncogenic *Kras* expression elevates acetyl-CoA availability in acinar cells, which causes an increase in histone acetylation [64]. Targeting acetyl-CoA generation suppresses histone hyperacetylation and blocks duct formation in vitro and in vivo [64]. 

It is well appreciated today that cells massively rewire their cellular metabolism to enhance plasticity, largely but not exclusively by impacting the epigenome [97]. Over the past two decades, a compelling body of work has shown that metabolism is profoundly altered in PDA, unveiling vulnerabilities and translational opportunities [98]. It is intuitive to think that acinar-to-ductal metaplasia may also demand very specific energetic and anaplerotic rearrangements (Figure 4B), which are however extremely challenging to study in vivo due to the scattered nature of intra-epithelial lesions. Instead, in vitro systems are well suited for the interrogation of labile metabolites which intracellular levels fluctuate very rapidly and vary significantly across cell types. Thus, well-defined cell-based approaches are usually preferred for metabolites analysis. In addition, cell culturing allows the study of metabolic fluxes, where the fate of stable isotope-labelled nutrients (usually carbon-13, ^13^C) can be traced using mass spectrometry, providing a more dynamic picture of the metabolic activity of the cell [99,100].

Interestingly, flux analysis on purified acini revealed that an increased fraction of cytosolic acetyl-CoA is also channeled toward the synthesis of cholesterol during ex vivo ADM [64]. While this and other converging evidence suggest that sterols might play a crucial role in pancreatic tumorigenesis [101,102], the exact mechanism is still elusive. 

Unhealthy dietary habits are linked to increased risk of pancreatic cancer [3], which highlights the need to clarify how nutrients are utilized by premalignant acinar cells. ^13^C tracing experiments in isolated acini indicated that branch chain amino acids (BCAAs) are a preferred carbon source over glucose and fatty acids [64]. Interestingly, BCAAs are oxidized in the tri-carboxylic acid (TCA) cycle [103,104] and support acetyl-CoA generation and histone acetylation in acinar cells. Targeting BCAA catabolism in ex vivo cultures suppresses acinar-to-ductal metaplasia [104]. 

Like most oncogenes, mutant *Kras* actively rewires cellular metabolism to make cells thriving in harsh environments. Mitochondria are the pivotal hubs that control energy production, redox balance and several anaplerotic pathways, while also integrate signaling events with nutrient sensing and catabolism [105]. Mitochondria are reprogrammed in PDA cancer cells and directly impacted by KRAS^G12D^ [106,107,108,109]. This has been shown in multiple studies using organotypic cultures. First, ex vivo activation of the mutant KRAS allele leads to impaired mitochondrial respiration, accompanied by the generation of reactive oxygen species (ROS), whereas quenching of ROS suppresses ADM in vitro [50]. Second, mitochondria hyperpolarization contributes to acinar cell dysfunction in CCK-induced pancreatitis [110]. Third, converging evidence suggests that changes in mitochondrial morphology, which influence their function, critically support PDA tumorigenesis [108,111,112], although the impact of mitochondrial dynamics on acinar cell transformation has not directly been tested yet. In addition, it is plausible that the reprogramming of mitochondrial morphology and function influences Ca^2+^ homeostasis that controls several aspects of acinar cell pathophysiology [10].

Overall, these data are consistent with the hypothesis that a switch to a ductal-like phenotype is associated and supported by extensive reprogramming of cellular metabolism. The complexity of catabolic and anabolic processes, the bi-directionality of many reactions and the intrinsic flexibility of metabolism pose significant technological and conceptual challenges to the screening of ADM-supporting metabolites. However, modern metabolomics approaches are being applied to acinar cell cultures in order to fully uncover oncogene-induced, PDA-initiating metabolic rewiring.

### 6.5. Non-Cell Autonomous Stimulation of Cultured ACs

Both physiological and pathological acinar metaplasia are critically regulated by stromal accessory cells [22]. In mice, the injection of cerulein, a CCK analog, induces a strong pancreatitis associated with the recruitment of a massive immune infiltrate and diffuse acinar metaplasia [21,64,81,82]. Although the isolation and culture of primary acini is primarily used to uncover cell-autonomous mechanisms that regulate cell transdifferentiation, ad hoc experiments can also be useful to define the paracrine contribution of secrete factors. In this way, the Storz group was able to dissect the contribution of M1 macrophage-secreted tumor necrosis factor alpha (TNFα, among other cytokines) and matrix metalloproteases to inflammation-induced ADM [73,113].

Paracrine activity can also restrict ADM. Primary fibroblast-conditioned medium inhibits metaplasia of purified acini, an effect dependent on intact Hedgehog signaling [114]. One could envision a model where cell plasticity is in fact continuously restrained by local factors and normal pancreatic tissue architecture. In that case, the isolation of acinar cells might remove environmental brakes and unleash their intrinsic plasticity. In line with this hypothesis, it has been demonstrated that isolated acinar cells acquire stem cell-like traits that can be transmitted through multiple passages [115].

### 6.6. Ex Vivo Modeling of Cell Extrinsic Constrains of Acinar Cell Plasticity 

One can reason that preservation of pancreatic lobular architecture is critical to constrain acinar cell plasticity. This hints at the possibility that mechanosensing mechanisms may play a role in ADM. The Hippo pathway and its terminal transcriptional effectors YAP and TAZ, transduce the sensing of structural and mechanical cues of the microenvironment [116]. Stiff, tumor-like matrices (such as Matrigel^®^) induce nuclear relocation of YAP, which activates the Hippo signaling and trigger ADM [117]. YAP/TAZ signaling is activated by malignant extrinsic cues and is implicated in the initiation of solid tumors [116]. YAP and TAZ are upregulated in pancreatic acinar cells following injection of cerulein in KC mice [118], which creates a swollen and stiff local environment. Overexpression of either YAP or TAZ in isolated primary acini accelerates ductal transition, while *Cre*-mediated gene ablation preserve acinar morphology [115,118]. 

Organ structure is also maintained by homeostatic mechanisms. Tissue homeostasis is the primary control of malignant transformation and cancer initiation [119]. In fact, maintenance of genetic and functional integrity is achieved through the turnover of senescent or damaged cells, which is largely mediated by programmed cell death in post-mitotic tissues like the pancreas [119]. It is tempting to speculate that apoptosis may protect from irreversible ADM. In line with this hypothesis, members of the pro-apoptotic heat-shock protein (HSP) family have been implicated in pancreatic tumor development [67,120]. Also, hyperthermia induces HSP70 expression and protects pancreatic architecture after cerulein-induced inflammation [121]. Similarly, sodium arsenite-mediated upregulation of HSP70 and HSP27 preserved the function and structure of purified acini [122]. 

Finally, vagal innervation provides parasympathetic inputs that prevent PDA onset [123]. Administration of the muscarinic agonist bethanechol to ex vivo acinar explants blocks ADM, while vagotomy suppresses tumor formation in vivo in KC mice [123]. This is part of an emerging body of evidence attesting to a neural influence on cancer development and progression.

Collectively, these data point to the existence of in vivo constraints that limit the natural plasticity of acinar cells and show that these can be efficiently interrogated also using ex vivo cultures of acinar cells.

## 7. Conclusions

Acinar-to-ductal metaplasia is widespread in KC mice and gives origin to most precancerous lesions in model organisms. Isolation of primary acini and their transdifferentiation in vitro is a useful tool to decipher the network of events that lead to this permanent switch of cellular identity, which marks the initial step in pancreatic carcinogenesis. Similar to many other in vitro systems, it offers several advantages, including the possibility to perform large genetic or compound screening in a controlled genetic background. It permits the study of the transdifferentiation process live while it happens in the plate. In addition, metabolic and epigenetic analyses for pancreatic pre-neoplastic cells would be extremely challenging in in vivo settings, and the heterogeneous nature of the pancreatic parenchyma limits the value of whole-organ examinations. 

In contrast, major limitations inherent to the 3D culture of acinar cells are the difficulty to extend the culture for a period of time long enough to allow a long-term experiment and the significant stress and variability introduced by isolation procedures.

It is increasingly clear that pancreatic acinar cells are a diverse bunch [11,12,13]. It is conceivable that one or more subsets of acinar cells might be endowed with superior tumor-initiation capacity; for example, FACS-sorted Doublecortin-like Kinase *1* (*Dlck1*)-expressing acinar cells (represent 0.1–0.5% of the exocrine pancreas) more potently generate duct-like spheres in vitro [124]. At the same time, regional differences in vivo (nutrient/oxygen supply, distance from hormone-producing cells) might impact acinar cell phenotype. These levels of complexity cannot be recapitulated ex vivo through primary culture of acinar cells, as traditionally performed. Yet, the combination of organotypic culture with the application of subset tracing using genetically-encoded fluorophores, including multicolor technology, may be regarded as interesting routes of investigation [124,125].

Tumorigenic driver mutations are prevalent in humans, yet only very few cells transform. In KC mice, oncogenic *Kras* expression is widespread in the pancreatic epithelium but only a few scattered pre-neoplastic lesions arise. What renders a subset of acinar cells susceptible to transformation while the remaining population appears refractory? This is a fundamental question in cancer biology. Notably, ex vivo cultures are valuable tools to address this question in PDA. Flow cytometry of cultured acinar cells may permit to discriminate and separate cells undergoing metaplasia in response to certain stimuli, from cells undergoing incomplete transdifferentiation or maintaining their acinar phenotype. Our laboratory is interested in metabolic and epigenetic factors that facilitate oncogenic transformation of acinar cells.

The pivotal role of acinar cells in pancreatic cancer initiation has been outlined [22] (Figure 3). However, several aspects of acinar cell biology remain elusive, including the similarities (or lack of) between mouse and human pathophysiology. Human pancreatic explants are scarce, as pancreas biopsies are rarely performed due to the secluded anatomical location and no cell lines have been established, due to their high specialization and reliance on 3D organization. Human-derived acinar cells are becoming increasingly available as a result of the growing popularity of Langerhans’ islet transplantation procedures from donor pancreata, the leftovers of which can be purified and enriched for the acinar cell fraction [41]. On the contrary, primary acini are usually isolated from rodents that are conveniently available in most laboratories. It is fair to ask to what extent results can be translated to the human disease. The observation that human acinar explants undergo permanent ductal transdifferentiation in response to oncogenic stimuli and MAPK signaling [41] supports the hypothesis that mechanisms of PDA initiation could be conserved from mouse to humans, at least in part. In addition, proteomic and functional analyses of human acini extracted from cadavers showed secretory and inflammatory responses very similar to those observed in rodents [126]. However, human cells do not express the receptor for Cholecystokinin-A (*Cckr1*) and are thus weakly responsive to CCK stimulation [126]. This highlights that mouse and human cells do differ in certain aspects, and a potential species-specific effect should be taken into consideration. 

Improved prevention strategies have been effective in curtailing disease incidence and mortality over the past 20 years. In contrast to overall advances in cancer treatment and care, PDA incidence is rising, as well as the toll of annual deaths, while its mortality rate is still abysmal. A better understanding of the molecular steps leading to disease inception is critical to refine current prevention strategies and reduce the number of new cases, which appear paramount for a disease typically diagnosed late and lacking effective therapies. The singularity of acinar cells, which constitute a highly typified cell population, demands a dedicated in vitro system for the study of cell-specific events that cause neoplastic transformation.

## Figures and Tables

**Figure 1 cancers-12-02606-f001:**
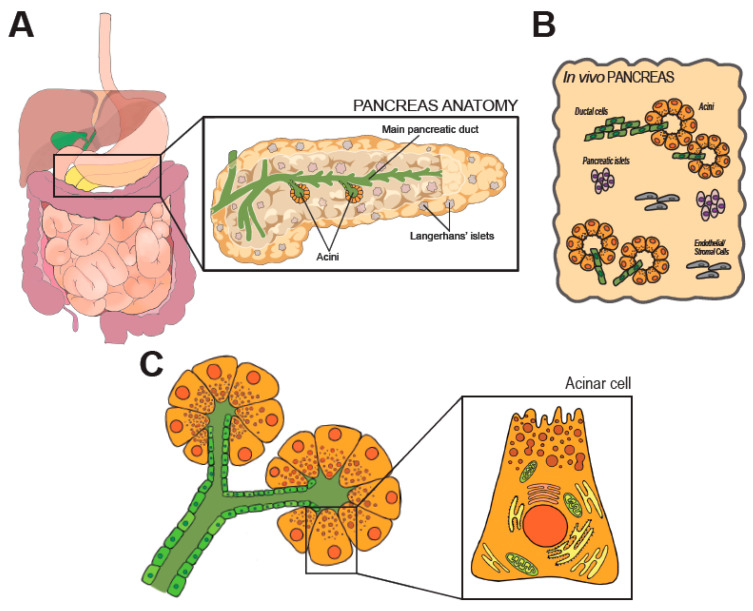
Anatomy and histology of the pancreas. In (**A**) anatomic organization of the GI tract. The pancreas is magnified on the right, along few representative structures within the organ. (**B**) shows the cellular heterogeneity of the pancreas, which is a composite of exocrine (acini), endocrine (Langherans’ islets) and stromal elements. (**C**) shows the modular units of the exocrine pancreas (acini). Morphology of an acinar cell is magnified on the right.

**Figure 2 cancers-12-02606-f002:**
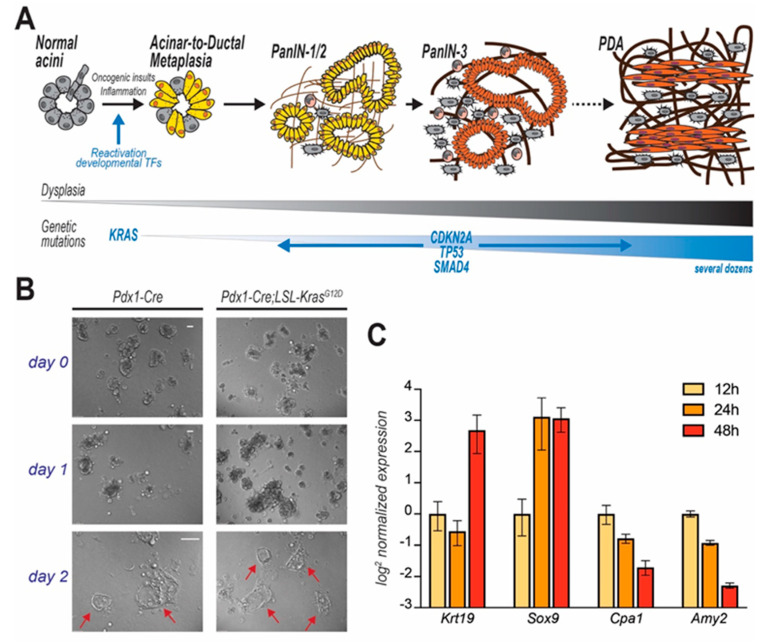
PDA progression and in vitro modelling of disease initiation. In (**A**) step-wise model of PDA progression. Upon mutation of KRAS, pancreatic acinar cells undergo acinar-to-ductal metaplasia (ADM), a cell transforming event which initiates the carcinogenic cascade. Cells progressively acquire dysplastic features and accrue genetic mutations. Acinar cell-derived pre-cancerous lesions are termed Pancreatic Intraepithelial Neoplasms (PanIN) that are defined by different histological grades and eventually evolve to carcinomas. In (**B**) acinar cells isolated from mice (genotypes indicated), immediately embedded in Matrigel and observed over time. Duct-like cystic structures (arrows) are observed after few days. In (**C**) time course analysis reveals that KC-derived primary acinar cells undergoing ADM ex vivo (Matrigel-embedded) upregulate duct-specific genes (*Krt19* and *Sox9*), and suppress acinar-specific genes (*Cpa1* and *Amy2*). Data for each gene are normalized to the first time point (12 h post isolation).

**Figure 3 cancers-12-02606-f003:**
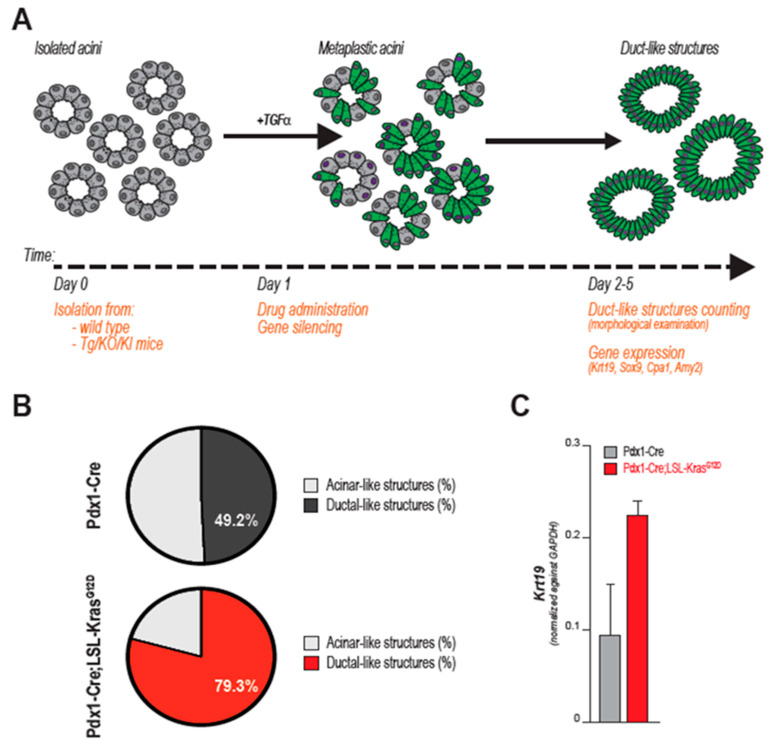
Ex vivo acinar-to-ductal metaplasia (ADM) as a model for PDA initiation. In (**A**) schematic representation of ADM from acinar explants. Once isolated, acinar cells undergo transdifferentiation to acquire morphological and molecular features typical of ductal cells. Gene mutations (i.e., *KRAS*) and growth factors (i.e., TGFα, EGF) induce ADM ex vivo. The impact of additional genetic aberrations, signaling events or chemical compounds can be examined by several means (listed in orange): the ability to form ductal-like structure can be evaluated at end point, as shown in B and C. In this way, impact for tumor initiation can be inferred. In (**B**,**C**), two different metrics to score the tumor-initiating potential of *KRAS* mutation. In (**B**) percentage of acinar- and ductal-like structures is assessed by morphological evaluation of multiple samples. Acinar cells derived from *Pdx1-Cre*; *LSL-KrasG12D* (KC) mice form significantly more duct when cultured ex vivo (day 2 post isolation, in Matrigel). Percentage in white indicates the fraction of duct-like structures arising from cultures of acinar cells derived from either *Pdx1-Cre* or KC mice (on the total number of multi-cellular structures analyzed). In (**C**) quantitative PCR shows increased expression of duct-specific Cytokeratin-19 (*Krt19*) in cultures derived from either *Pdx1-Cre* or KC pancreata (at day 2 post isolation, in Matrigel, triplicates).

**Figure 4 cancers-12-02606-f004:**
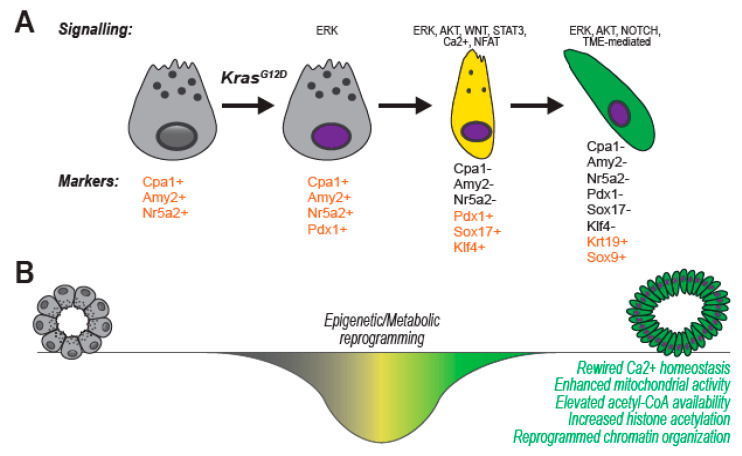
Cellular reprogramming during acinar-to-ductal metaplasia (ADM). In (**A**) ex vivo-isolated acinar cells transition to a tumor-initiating ductal phenotype. Cell identity is impacted: undergoing ADM, the cell loses its secretory function (apical granules disappear), changes morphology (acquire a more elongated, brick-like structure). Gene expression and markers are dynamically regulated, as shown in the lower panel. Transition in cell identity is a continuum, which progresses through de-differentiation into a progenitor-like cell type (in yellow) and culminates in a duct-like cell (in green). (**B**) Acinar-to-Ductal Metaplasia is critically supported by both epigenetic and metabolic reprogramming. Mechanisms for active cellular remodeling are necessary to switch cell identity across two terminally-differentiated states.

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
