# Peer review of "Organotypic Culture of Acinar Cells for the Study of Pancreatic Cancer Initiation"

_cancers, 2020, doi:10.3390/cancers12092606_

Round 1

Reviewer 1 Report

The review article by Carlotta Paoli and Alessandro Carrer is of interest to those interested in the study of pancreatic cancer carcinogenesis. The review discusses the isolation and culture of primary mouse pancreatic acinar cells, providing and historical and technical perspective.There are no revisions needed for this reviewer.

Author Response

Most significant changes to the revised manuscript are as follows:

  • The abstract has been edited to convey the key message more directly, as Reviewers noted a lack of clarity.
  • A new Figure 3 was designed. It shows schematically how ex vivo culture can be used to quantify tumorigenic potential of gene mutations, or other genetic and/or chemical manipulations. Oncogenic mutation of KRAS was used as example. As result, the manuscript now includes 4 Figures.
  • A new section was added at the end of Chapter 6, as recommended by Reviewer #3. The section discusses microenvironmental constraints of acinar cell plasticity. Data presented highlight the roles of mechano sensing (previously in the “signaling” section), tissue homeostasis and apoptosis (as suggested by Reviewer #2) and influence of local innervation.

Please find below a point-by-point reply to the Reviewers’ comments.

Reviewer #1

The review article by Carlotta Paoli and Alessandro Carrer is of interest to those interested in the study of pancreatic cancer carcinogenesis. The review discusses the isolation and culture of primary mouse pancreatic acinar cells, providing and historical and technical perspective.
There are no revisions needed for this reviewer.

Reply: We thank the Reviewer. The review is indeed thought to be of interest for the pancreatic cancer community.

Reviewer 2 Report

In this interesting review authors present the role of isolated acinar cell culture as the model to study pancreatic cancerogenesis. In vitro spheroids of acinar cells undewent transformation acinar to ductal neoplasia, under special conditions. Authors discussed the mechanisms of  pancreatic oncogenic transformation relative to genetic mutation with special role of KRAS signaling pathways.

The manuscript characterized the molecular mechanisms, which are believed to be responsible for pancreatic oncogenesis, including factors affecting intracellular signaling pathways, post-translational  modifications of DNA and histones and changes of cellular metabolism. The incorporated figures illustrate the thesis  presented in this article.

Minor points:

  1. It should be mentioned in the manuscript that for in vitro experiments acinar cells obtained from the rat pancreas, were commonly used, together with these from guinea pig, and later – from mouse. Acini obtained from species different from human are characterized by presence of different receptors (f.ex receptors for secretin and CCKR1 are present on rat pancreas, not on human or dog’s cells).
  2. To prevent autodigestion of acinar cells inhibitor of serine-treonine proteases (trasylol/aprotynin) was added to the incubation  medium of pancreatic acini (method of Amsterdam and Jamieson with modification).
  3. The role of apoptosis in the prevention of pancreatic cancer development should be also mentioned, as demonstrated in the previous publications on pancreatic tumor cell line. Inhibition of pro-apoptotic signaling pathway by heat shock proteins could be implicated in the pancreatic tumorigenesis (Pancreatology 2017,  J.Physiol. Pharmacol 2015).  
  4. Addition of abbreviations list in the end of manuscript should be very helpful for readers to understand the problems of molecular and intracellular changes leading to oncogenic transformation. By the way - abbreviations MEK (MAPK/ERK) and ECM (extracellular matrix) are not explained in the text.

Author Response

Most significant changes to the revised manuscript are as follows:

  • The abstract has been edited to convey the key message more directly, as Reviewers noted a lack of clarity.
  • A new Figure 3 was designed. It shows schematically how ex vivo culture can be used to quantify tumorigenic potential of gene mutations, or other genetic and/or chemical manipulations. Oncogenic mutation of KRAS was used as example. As result, the manuscript now includes 4 Figures.
  • A new section was added at the end of Chapter 6, as recommended by Reviewer #3. The section discusses microenvironmental constraints of acinar cell plasticity. Data presented highlight the roles of mechano sensing (previously in the “signaling” section), tissue homeostasis and apoptosis (as suggested by Reviewer #2) and influence of local innervation.

Please find below a point-by-point reply to the Reviewers’ comments.

Reviewer #2

In this interesting review authors present the role of isolated acinar cell culture as the model to study pancreatic cancerogenesis. In vitro spheroids of acinar cells undewent transformation acinar to ductal neoplasia, under special conditions. Authors discussed the mechanisms of  pancreatic oncogenic transformation relative to genetic mutation with special role of KRAS signaling pathways.

The manuscript characterized the molecular mechanisms, which are believed to be responsible for pancreatic oncogenesis, including factors affecting intracellular signaling pathways, post-translational  modifications of DNA and histones and changes of cellular metabolism. The incorporated figures illustrate the thesis  presented in this article.

Reply: We thank the Reviewer and appreciate her/his positive feedback.

Minor points:

  1. It should be mentioned in the manuscript that for in vitro experiments acinar cells obtained from the rat pancreas, were commonly used, together with these from guinea pig, and later – from mouse. Acini obtained from species different from human are characterized by presence of different receptors (f.ex receptors for secretin and CCKR1 are present on rat pancreas, not on human or dog’s cells).

Reply: Reviewer’s point is well taken. In the original manuscript, we did touch on similarities between human and mouse acinar cells in the “Conclusion” section. We decided to add comments on receptor expression, responsiveness to CCK and also acknowledge critical differences between mouse and human cells (503-512).

In addition, we more clearly stated in the text that rodents’ cells are typically used for experimentation. Hence, potential differences have to be taken into account (195-196).

We believe this addresses Reviewer’s comment.

  1. To prevent autodigestion of acinar cells inhibitor of serine-treonine proteases (trasylol/aprotynin) was added to the incubation  medium of pancreatic acini (method of Amsterdam and Jamieson with modification).

Reply: This point is absolutely important, and it has been added to the manuscript (162-164).

  1. The role of apoptosis in the prevention of pancreatic cancer development should be also mentioned, as demonstrated in the previous publications on pancreatic tumor cell line. Inhibition of pro-apoptotic signaling pathway by heat shock proteins could be implicated in the pancreatic tumorigenesis (Pancreatology 2017,  J.Physiol. Pharmacol 2015).  

Reply: We agreed that constraints of apoptosis are important tumorigenic mechanisms. Although they have been extensively studied in tumor cells and are well-appreciated mechanisms of immortalization and senescence evasion, no specific experiments have been performed on primary acinar cells undergoing metaplasia, to our knowledge.

Although we could not track down the specific papers mentioned by the Reviewer, we do acknowledge there is significant evidence for a) heat shock proteins and apoptosis in pancreatic tumor cells, b) heat shock proteins in pancreatitis and c) apoptosis in acinar cell loss following pancreatitis. Collectively, those provide substantial (although indirect) evidence for Reviewer’s hypothesis. We decided to mention the roles of apoptosis and HSP70 in a new section dedicated to tissue homeostasis and microenvironmental restraints of ADM; specifically, lines 448 to 456. Our writing may be a bit speculative for the lack of experiments on acinar cells, but we hope this might suite well Reviewer’s issue. Our deepest apologies for possibly missing out some important literature related to the topic.

  1. Addition of abbreviations list in the end of manuscript should be very helpful for readers to understand the problems of molecular and intracellular changes leading to oncogenic transformation. By the way - abbreviations MEK (MAPK/ERK) and ECM (extracellular matrix) are not explained in the text.

Reply: MEK acronym has been spelled out, while ECM substituted with “extra-cellular”. A list with frequently used abbreviations has been added at the end of the manuscript.

Reviewer 3 Report

Pancreatic Ductal Adenocarcinoma (PDA) has very low survival rate mostly because it is often diagnosed late, and very little is known about the mechanisms underlying it's onset. Mounting recent studies strongly suggest that despite the duct-like appearance PDA originate from pancreatic acinar cells. Yet, very little is known of the initial transformative steps of acinar cells to PDA. This review article highlights the importance of organotypic acinar culture for understanding the initial mechanistic steps underlying the transformation of acinar cells into PDA. The authors offer a comprehensive examination of the unique advantages of 3D acinar cell culture in providing insights into the initial events of acinar cell transformation. However in the current state the manuscript is poorly organized making it is extremely challenging for a reader to reap the benefits of this, otherwise, very informative review article.

It is the view of this reviewer that, the authors need to reorganize the manuscript to offer clarity, and to make this review a valuable resource for the research community. Below are specific comments for changes.

Major comments:

1) Given that both wild type and mutant acinar cells can spontaneously under ADM, what is the most definitive metric for determining which cell is most tumorigenic in vitro?

It appears in the current manuscript that both wild type and Kras expressing acinar cells would both spontaneously undergo ADM in vitro. However in vivo only the latter would undergo ADM. Therefore, it is not clear in the current manuscript, how current organotypic culture of acinar cells serves as a definitive assay for assessing the tumorigenic potential of acinar cells in vitro. The authors should expound and clarify this, as this forms the central theme of the review.  For example in Fig 2B, both Pdx1-Cre and Pdx1-Cre;LSL-KrasG12D form ADM at day2 in matrigel. Also Fig2 C, (which is presumably wild type acinar cells) lose acinar makers and gain duct markers over time. How may a reader of this manuscript, definitively assess between two acinar cells of distinct genotypes, which one is more tumorigenic, using in vitro assay? On the other hand in reference [46], which is based on acinar cells cultured in collagen, it seems wild type acinar cells do not survive, whereas Kras expressing cells undergo ADM. Is this right? Please clarify and expound on this point.

If both wild type and mutant acinar cells can spontaneously undergo ADM, is the metric for determining which is most tumorigenic based solely on level of expression of duct markers, as shown in Fig 2B?

2) Directly related to the above, it would be helpful if the authors dedicate a separate section to discuss whether local factors in vivo retrain ADM (This is mentioned briefly in 431-432). And if so, is it the absence of such restraining factors that leads to spontaneous ADM in vitro? And which culture conditions (if know) can mimic the in vivo setting. If not currently known, then the authors should highlight that as a current limitation of in vitro culture of acinar cells. This issue can be discussed together with the effect of matrigel on ADM, and whether it is due to stiffness and thus Hippo signaling or growth factors from matrigel. This would be easier for a reader than transitioning from MAPK signaling to the effect of matrigel, as is currently written.

3) The following statements appear to contradict each other:

(i) 209-211: Although cell transdifferentiation occurs spontaneously in these settings, single-allelic Kras mutations significantly accelerate ADM and enhance the expression of ductal markers like Sox9 and Krt19 , while suppress canonical acinar genes (Mist1 , Amylase )”.

and,

241-243: Yet, converging observations suggest that only sustained levels of MAPK signaling activity are

per se able to trigger ADM and tumor initiation; whereas mono-allelic activating KRAS mutations might not be sufficient”

(ii) 285-286: “Reactivation of developmental programs augments cell plasticity: identifying critical hubs using primary acinar cultures”

and

“Aggregate evidence indicates that the expression of developmental regulators of acinar specification restrains cell identity during tissue injury/regeneration and opposes KrasG12D –induced tumorigenesis”

The former statement seems to suggest that developmental genes promote plasticity (plasticity here could mean potential for ADM), whereas the latter statement seems to suggest that developmental genes act to maintain cell identity and prevent Kras induced ADM and consequently tumorigenesis.

These discrepancies make the entire section incomprehensible.

Minor comments:

1) 390-391: “Future studies will test the value of cholesterol-abating interventions (e.g., statin administration, dietary restrictions) in preventing disease onset.” Are the authors suggesting to the research community that future studies should aim to address the above stated point? If so it is not clear as stated. On the other hand if they intend to suggest that they the authors plan to address this in future, that is not clear either.

2) The authors should state in Figure 2 legend, what day (0, 1 or 2) the qPCR graph was derived from. Also it should be clearly stated if Fig 2C is derived from wild type acinar cells or KC acinar cells.

3) The are numerous grammatical errors throughout the manuscript that makes it challenging to comprehend what would otherwise be a valuable manuscript for the field.

Author Response

Most significant changes to the revised manuscript are as follows:

  • The abstract has been edited to convey the key message more directly, as Reviewers noted a lack of clarity.
  • A new Figure 3 was designed. It shows schematically how ex vivo culture can be used to quantify tumorigenic potential of gene mutations, or other genetic and/or chemical manipulations. Oncogenic mutation of KRAS was used as example. As result, the manuscript now includes 4 Figures.
  • A new section was added at the end of Chapter 6, as recommended by Reviewer #3. The section discusses microenvironmental constraints of acinar cell plasticity. Data presented highlight the roles of mechano sensing (previously in the “signaling” section), tissue homeostasis and apoptosis (as suggested by Reviewer #2) and influence of local innervation.

Please find below a point-by-point reply to the Reviewers’ comments.

Reviewer #3

Pancreatic Ductal Adenocarcinoma (PDA) has very low survival rate mostly because it is often diagnosed late, and very little is known about the mechanisms underlying it's onset. Mounting recent studies strongly suggest that despite the duct-like appearance PDA originate from pancreatic acinar cells. Yet, very little is known of the initial transformative steps of acinar cells to PDA. This review article highlights the importance of organotypic acinar culture for understanding the initial mechanistic steps underlying the transformation of acinar cells into PDA. The authors offer a comprehensive examination of the unique advantages of 3D acinar cell culture in providing insights into the initial events of acinar cell transformation. However in the current state the manuscript is poorly organized making it is extremely challenging for a reader to reap the benefits of this, otherwise, very informative review article.

It is the view of this reviewer that, the authors need to reorganize the manuscript to offer clarity, and to make this review a valuable resource for the research community. Below are specific comments for changes.

Reply: We thank the Reviewer for the careful reading and insightful comments. We thought all her/his points are well taken; addressing them we could significantly improve the manuscript. See below for detailed considerations and specifics of our editing. We appreciate (s)he also thinks highly of the potential value of this review article for the pancreatic cancer community, and acknowledges scarce information on initial transformative events on acinar cells.

Major comments:

1) Given that both wild type and mutant acinar cells can spontaneously under ADM, what is the most definitive metric for determining which cell is most tumorigenic in vitro?

It appears in the current manuscript that both wild type and Kras expressing acinar cells would both spontaneously undergo ADM in vitro. However in vivo only the latter would undergo ADM. Therefore, it is not clear in the current manuscript, how current organotypic culture of acinar cells serves as a definitive assay for assessing the tumorigenic potential of acinar cells in vitro. The authors should expound and clarify this, as this forms the central theme of the review. 

Reply: As the Reviewer said, this is the central point of the manuscript. To address her/his concern, we extensively edited the manuscript at several levels to improve clarity and deliver more effectively:

  • We significantly re-wrote the abstract, to make it more straightforward and to include a quick take on acinar-to-ductal metaplasia and its role in tumorigenesis.
  • We created an additional Figure (the newly designed Figure 3) dedicated to the visual explanation of the ADM model. It illustrates that either morphological examination (using a simple optic microscope) or quantitative PCR of ductal genes is used to assess metaplasia and to measure the tumorigenic potential of genetic or signaling alterations.
  • To the same aim, we implemented Chapter 5 to include a more step-wise description of ex vivo acinar-to-ductal metaplasia as a tool to study tumor initiation. Methods used to quantify tumorigenic potential of genetic modification or drug administration have been also better outlined. See lines 196-214.
  • We re-arranged the “Pancreatic carcinogenesis” section to highlight the role of ADM in physiology (102-108). This also explains that wild-type acinar cells do undergo spontaneous metaplasia both in vitro and in vivo, which seemed a point of confusion.
  • The second part of the newly designed Figure 3 shows how oncogenic events impact ADM ex vivo. Increase in the number of duct-like structures in the plate, and elevation of the expression of duct-specific genes are observed. The case of oncogenic KRAS mutations is reported. We chose this example because KRASG12D is the most well-characterized transformative alteration for pancreatic epithelial cells. This illustrative example can be extended to all sorts of in vitro/ex vivo
  • We took the liberty to further interpret Reviewer’s thinking. “What makes a cell susceptible to oncogenic transformation?” is an outstanding and yet unsolved question in cancer research. We briefly discussed this aspect in the conclusions (lines 487-495). Ex vivo organotypical cultures may actually represent an exciting tool to address this question, because it is technically easier to discriminate between cells undergoing full ADM, incomplete/partial ADM, or not undergoing ADM at all.

For example in Fig 2B, both Pdx1-Cre and Pdx1-Cre;LSL-KrasG12D form ADM at day2 in matrigel.

Also Fig2 C, (which is presumably wild type acinar cells) lose acinar makers and gain duct markers over time. How may a reader of this manuscript, definitively assess between two acinar cells of distinct genotypes, which one is more tumorigenic, using in vitro assay?

Reply: As mentioned, the issue has been addresses at multiple levels in the text. Briefly, the genotype that forms more ducts in culture is more tumorigenic.

With regard to the data showed in the Figures, wild-type acinar cells do undergo spontaneous metaplasia in Matrigel, likely due to reasons described in the manuscript. As exemplified in the new Figure 3 (which includes the data previously shown in Figure 2B), oncogenic inputs augment the number of duct-like structures at end point: this readout is quantified by counting the number of duct-like structures at the microscope, or measuring the expression of duct-specific genes by qPCR. If cells have been manipulated, results are correlated to the tumorigenic properties of the manipulation.

As for Figure 2C, cells are derived from KC mice. This is now reported in the figure legend.

On the other hand in reference [46], which is based on acinar cells cultured in collagen, it seems wild type acinar cells do not survive, whereas Kras expressing cells undergo ADM. Is this right? Please clarify and expound on this point.

Reply: In collagen, acinar cells are not exposed to inputs to undergo ADM, unless either TGFa or EGF is added. They do survive for a defined period of time, but as most post-mitotitc cells in culture they eventually die. In contrast, Matrigel contains a plethora of growth factors that spontaneously trigger ADM without the need for TGFa/EGF supplementation. This is now better explained in 199-201, and comprehensively addressed in the text as described above.

If both wild type and mutant acinar cells can spontaneously undergo ADM, is the metric for determining which is most tumorigenic based solely on level of expression of duct markers, as shown in Fig 2B?

Reply: qPCR and morphological examination. Results are usually superimposable. The process is now better explained at several levels, as described above.

2) Directly related to the above, it would be helpful if the authors dedicate a separate section to discuss whether local factors in vivo retrain ADM (This is mentioned briefly in 431-432). And if so, is it the absence of such restraining factors that leads to spontaneous ADM in vitro? And which culture conditions (if know) can mimic the in vivo setting. If not currently known, then the authors should highlight that as a current limitation of in vitro culture of acinar cells. This issue can be discussed together with the effect of matrigel on ADM, and whether it is due to stiffness and thus Hippo signaling or growth factors from matrigel. This would be easier for a reader than transitioning from MAPK signaling to the effect of matrigel, as is currently written.

Reply: We believe this is a great suggestion. We did create a dedicated section entitled “Ex vivo modeling of cell extrinsic constraints of acinar cell plasticity” (437-463), where we outline data and hypotheses suggesting the existence of in vivo constraints that limit the intrinsic plasticity acinar cells are endowed with. The section also integrates the role of mechano sensing. The underlying hypothesis is that the physiological tissue architecture and homeostasis preserve the identity of pancreatic cell types. We think the new structure improved the flow of the manuscript and satisfies Reviewer’s issue.

While we think the data and hypothesis outlined are exciting, we acknowledge that this is an aspect that has not been explored before and only scant evidence is available. We thank the Reviewer for the opportunity to summarize what is currently known and put forward an unprecedented hypothesis (to our knowledge).

3) The following statements appear to contradict each other:

(i) 209-211: Although cell transdifferentiation occurs spontaneously in these settings, single-allelic Kras mutations significantly accelerate ADM and enhance the expression of ductal markers like Sox9 and Krt19 , while suppress canonical acinar genes (Mist1 , Amylase )”.

and,

241-243: Yet, converging observations suggest that only sustained levels of MAPK signaling activity are

per se able to trigger ADM and tumor initiation; whereas mono-allelic activating KRAS mutations might not be sufficient”

Reply: We removed second part, which was not integral to the narrative of the manuscript.

(ii) 285-286: “Reactivation of developmental programs augments cell plasticity: identifying critical hubs using primary acinar cultures”

and

“Aggregate evidence indicates that the expression of developmental regulators of acinar specification restrains cell identity during tissue injury/regeneration and opposes KrasG12D –induced tumorigenesis”

The former statement seems to suggest that developmental genes promote plasticity (plasticity here could mean potential for ADM), whereas the latter statement seems to suggest that developmental genes act to maintain cell identity and prevent Kras induced ADM and consequently tumorigenesis.

These discrepancies make the entire section incomprehensible.

Reply: The reviewer is right. While reactivation of certain factors in adult acinar cells is often associated with elevated plasticity, other developmentally-regulated transcription factors are obviously involved in acinar cell specification during embryogenesis and their uninterrupted expression preserve acinar cell identity and restrain plasticity in adulthood. We acknowledge that “developmental programs” is a wide umbrella definition, which may be too vague, and the paragraph title is misleading in its original form.

We replaced it with: “Master transcription factors shape cell identity: discerning critical hubs using primary acinar cultures”. 

We apologize for the lack of clarity. By making the title more crisp, we believe the entire section may be more readable. However, we introduced other notable changes to improve consistency and clarity in this section: we modified the introductory paragraph (290-293), added a brief paragraph on PDX1 (which marks early developmental progenitors; lines 301-303), and detailed the contrast with acinar specification genes (304-305).

Minor comments: 

1) 390-391: “Future studies will test the value of cholesterol-abating interventions (e.g., statin administration, dietary restrictions) in preventing disease onset.” Are the authors suggesting to the research community that future studies should aim to address the above stated point? If so it is not clear as stated. On the other hand if they intend to suggest that they the authors plan to address this in future, that is not clear either.

Reply: For the sake of clarity, we removed this sentence. The data that were actually discussed and the ultimate flow of the section are not impacted.

2) The authors should state in Figure 2 legend, what day (0, 1 or 2) the qPCR graph was derived from. Also it should be clearly stated if Fig 2C is derived from wild type acinar cells or KC acinar cells.

Reply: We fixed the figure legend. qPCR analysis in 2B (now in Figure 3) is relative to day 2 acinar cells. In 2C, acinar cells are derived from KC animals.

3) The are numerous grammatical errors throughout the manuscript that makes it challenging to comprehend what would otherwise be a valuable manuscript for the field.

Reply: We apologize for the poor grammar of the original manuscript. We had it checked by an English-speaking professional proofreader.

Round 2

Reviewer 3 Report

This manuscript has improved tremendously following the authors' revision. However, there a still some minor error and corrections that the authors should edit to improve the quality of the manuscript. 

Below are just a few examples:

Line 27: There is no apparent reason for abbreviating "INTRODUCTION"

Figure 2C in there is a typographical error "normalized" is misspelled.

Line 199: All subheadings in throughout the manuscript are in uppercase. Why is Line 199 in lower case?